# Ultra-low power, high-data rate, fully on-chip radio frequency on-off keying receiver for internet-of-things applications
Aasish Boora [1] ✉, Bharatha Kumar Thangarasu[1,2] & Kiat Seng Yeo[1,2]

Despite the enormous potential of energy-efficient receivers for wireless sensor networks, the large power consumption or limited data rate support impedes its extensive applications. Here, we present an energy-efficient, ultra-low power, higher data rate supporting, completely on-chip radio-frequency receiver frontend for on-off keying modulated signals in the 2.4 GHz industrial, scientific, and medical band. This compact-sized receiver is achieved by implementing temperature-resilient oscillator, pseudo-differential mixer, and a wideband detector while avoiding bulky external components such as bulk-acoustic wave resonators, crystal oscillators. Measurement results demonstrate that the proposed on-off keying receiver can decode low power level radio-frequency signals up to 5 Mbps data rate while consuming only 178 μW power. This work also demonstrates support for lower data rates at reduced power. Since the proposed receiver operates in different power modes, it can be integrated in diverse applications including internet-of-things devices and continuously monitoring biomedical/wearable implants.

Recent studies show that it is very important to achieve sustainable solutions in the domain of wireless communications[1,2]. Due to development of various technologies in the field of high-speed communications and artificial intelligence (AI) processing, the power consumption is rapidly growing. The most recent advancements in achieving low-power solutions include embedded sensor platforms[1], switching circuits[2], Si photonic circuits[3], and energy harvesting for internet-of-things (IoT) applications[4]. In IoT, data communication at a rapid pace is very important[5]. This demand comes from the immense scope of applications that could be addressed and fulfilled through some unique techniques in integrated circuit design[6]. For example, society sees many cases of fatality that could be prevented if only the cause was noticed a few seconds earlier. This often occurs in healthcare and other critical industries. Thus, it is important to establish a system that can detect errors within the least time after a trigger occurs[7]. Translating the scenario to wireless communications, it is desired to decode information with minimum latency. Not only is the latency that is important, but also the amount of information which is represented by the data rate of the signal[8]. Wireless sensor networks (WSNs) can benefit from high data rates to allow real-time decoding for environmental monitoring, industrial IoT, and smart agriculture. Healthcare monitoring devices benefit from a high data rate in

decoding vital signs through remote monitoring in real-time[9]. Other applications include smart home automation, asset tracking, inventory management, and cognitive cameras with high-volume data processing. The low-power wide-area network applications such as safety monitoring, smart metering, smart lighting, asset monitoring and tracking, and precision agriculture typically require a data rate of less than 100 kbps[10]. While the WSN applications such as cognitive surveillance, video streaming, and telemedicine usually require a data rate up to 4.5 Mbps[11] and minimal latency. The constraints on power consumption and area comes from a wide set of applications in WSNs including biomedical implants, remote sensing, and surveillance[12]. Bluetooth (BT) and wireless fidelity (Wi-Fi) systems are capable of handling higher data rates but will require complex modulation techniques that are power-hungry[13]. Unique design techniques must be incorporated in order to achieve high data rate support and low latency at lower power consumption and small area. It is essential to support high data-rate communication to facilitate and aid a wide range of applications. The proposed solution provides a holistic approach to address these sets of applications with minimal power consumption and low latency simultaneously. To prove the concept of the proposed receiver design methods, the on-off keying (OOK) modulation technique is chosen since it is one of the

[1]Singapore University of Technology and Design, Singapore 487372, Singapore. [2]Tianjin University, Tianjin 300072, China. ✉e-mail: rbna121197@gmail.com

most common modulation techniques prevailing in wake-up receivers (WuRx)[14–16]. The targeted specifications of this work are closely related to the wake-up receivers in wireless sensor node (WSN) applications.

From the state-of-the-art receivers, two receiver architectures namely "pre-amplified detector-first receiver architecture" and "mixer-first receiver architecture" are predominantly chosen to achieve the mentioned set of goals[17]. Benchmark designs have been published based on both these architectures with a substantial focus on low power consumption and better sensitivity using external components for input matching network (IMN), resonator[13,15–26]. Some designs have presented a very good sensitivity at high power consumption and decent data rates. Some designs have presented a good sensitivity at a very low power consumption but for very low data rates. Some other designs have presented a reasonable sensitivity at moderate power consumption and moderate data rates. This literature study suggests that a trade-off between sensitivity, power consumption, and data rate prevails in receivers. The upcoming IoT applications demand higher data rate support along with sufficient receiver sensitivity at a very low power consumption[14,17].

The detector-first receivers are mainly designed to achieve better sensitivity at ultra-low power consumption for low-data rate applications, but the envelope detector is a limitation for the receiver's ability to achieve high data rate support and good sensitivity due to inherent 1/f noise from the detector design[16]. To alleviate this issue and achieve better sensitivity of −64 dBm, the receiver is designed to support data rate OOK modulated signals at 100 kbps in the 2.4 GHz band. This supported data rate is a benchmark for WuRx with non-coherent detection architecture. Similar architecture with a common-source RF envelope detector is used to minimize the power consumption by 10-folds but achieved only a sensitivity of −50 dBm for a 200 kbps data rate[22].

On the other hand, the mixer-first receiver architecture has been chosen to achieve −72 dBm sensitivity for 100 kbps data rate applications at 52 µW power consumption[15]. An external bulk-acoustic wave (BAW) resonator is used to achieve the narrow RF filtering and 50 Ω input matching along with an on-chip capacitive transformer. A clock reference such as a 32 kHz real-time clock (RTC) is used to provide a reference signal to the frequency synthesizer to generate a stable local oscillator frequency[24]. External components are used for frequency synthesizer and input matching which occupies more area and requires a precise assembling process[24,27]. These designs have achieved a sensitivity of −72 dBm and −85 dBm at a power consumption of 95 µW and 220 µW for 62.5 kbps and 4 kbps, respectively. A few mixer-first receivers, which are designed to operate as a WuRx share the IMN with the actual receiver[27,28]. These existing receivers achieved improvement in the receiver sensitivity using external components and increased power consumption compared to the detector-first receivers[19,24,25,27,28] for a data rate of <10 kbps. Focussing on high data rates and low latency for low power consumption and reasonable sensitivity is essential.

There is a huge demand for compact and integrable features in the RF receiver, such as the 50 Ω RF front-end input matching and the baseband processor interfacing in IoT applications. To meet these requirements at very low-power dissipation, the RF receiver could be designed completely on a single die. This helps to minimize uncertainties such as external parasitics and RF front-end losses on the chip due to additional components for IMN. Thus, we proposed a completely on-chip ultra-low-power OOK receiver in 2.4 GHz ISM band for high data rate IoT and wireless sensor applications. In this work, a completely on-chip OOK receiver design in 40 nm CMOS technology is presented, which does not require any external components such as BAW resonators, crystal oscillators for integration with the receiver antenna. The proposed receiver design is fabricated in TSMC 40 nm low power (LP) mixed signal/radiofrequency (MS/RF) CMOS process. The semiconductor fabrication process comprises 10 metal layers for inter/intra-device connections. Different types of transistors such as regular-$V_T$, low-$V_T$, and high-$V_T$ devices are used in the receiver design. The high-frequency circuits are designed using high-frequency RF transistors from

the process design kit (PDK). These transistors possess a foundry-provided optimized layout to achieve optimized performance. The analog baseband circuits, such as the IF amplifier, envelope detector, comparator, and baseband amplifier, are designed using low-frequency transistors. The deep-well transistors are also available in the fabrication process, and are partly utilized in the foundry-provided input–output (IO) pad design. At each stage of the receiver design, appropriate transistors and other devices are chosen for optimum performance in the best knowledge of the authors. The miniature-sized receiver is achieved by considering a compact layout design and on-chip input matching compromising the additional RF front-end gain through external resonators in the matching network, which could improve the receiver sensitivity by at least 17–20 dB[15]. The proposed receiver is realized at a very low power consumption by considering a mixer-first receiver architecture as the LNA in the RF front-end could consume a large amount of power. In the proposed mixer-first receiver, a unique inverter-based active mixer topology is designed that consumes only 24 µW of power while providing an RF-to-DC conversion gain of more than 40 dB, including some passive RF voltage gain through the IMN. The proposed receiver also incorporates a stable on-chip local-oscillator (LO) signal generated from a modified voltage-controlled ring oscillator (VCRO) topology. This LO can be directly used in applications where the phase noise of the LO signal is not a major specification. There is also a provision to tune the frequency of the generated LO signal, which is briefly detailed in the methods section. The local oscillator generates a signal to down-convert the RF signal to an intermediate frequency (IF). A temperature-resilient current-starved VCRO is designed, which consumes very little power and aids the overall power budget of the receiver. The high data rate support at low power is achieved by using a wideband envelope detector. Thanks to the pseudo-differential characteristics of the mixer comprising single-ended RF and differential LO, which inherently generates a differential IF signal to the 5-transistor operational transconductance amplifier (5T-OTA) for additional gain. Thus, with the combination of an on-chip IMN, a pseudo-differential inverter-based active mixer, a temperature-resilient current-starved ring oscillator, and a wideband source-follower envelope detector, the proposed receiver is designed to occupy core die area of 0.122 mm² to support 5 Mbps/1 Mbps data rate with 295 ns latency at 178 µW/119 µW of power consumption achieving −62 dBm/ −67 dBm sensitivity.

## Results

The proposed ultra-low power receiver architecture and micro-photograph are shown in Fig. 1a and Fig. 1b, respectively. This receiver occupies a die area of 0.4 mm² with the probing pads included, and the circuit core dimensions are 370 µm × 330 µm as shown in Fig. 1b. The modes of operation of the proposed receiver, with supply voltages and current consumption details are provided in Table 1. The VCRO is operated at a voltage different from the rest of the blocks for two reasons. One reason is that to achieve the desired resilient square-wave signal at low power consumption, the supply voltage must be kept lower. Another reason is to have sufficient headroom for the $V_{DD}$ to tune and achieve the desired LO signal frequency.

All the DC supply voltages are provided using a Keysight B1500A semiconductor device analyzer, and the average current is sensed. For a better assessment of the proposed receiver, the input and output time-domain signal waveforms are monitored and recorded. A standard OOK modulated signal in the 2.4 GHz ISM band is provided to the receiver input using Keysight arbitrary waveform generator (M8190A). A Rohde & Schwarz oscilloscope (RTO1044) is used to monitor and record the receiver output by utilizing the high load impedance driving baseband amplifier. This equipment offers a load capacitance of 15 pF. The measurement setup schematic and the lab test bench are shown in Fig. 1c, d.

The input signal reflection parameter $S_{11}$ with post-layout simulation and measurement results is depicted in Fig. 2a. The sensitivity performance of the proposed receiver is determined as the input signal power level for

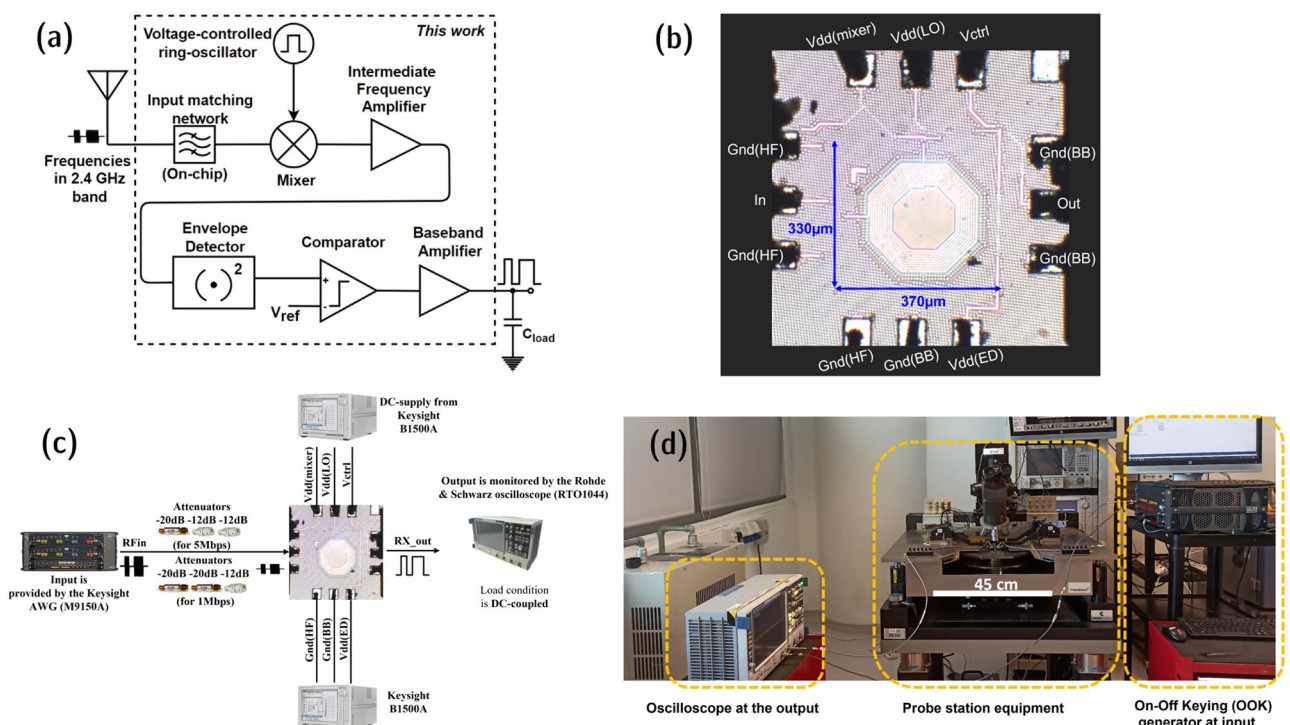

**Fig. 1 | Proposed receiver design and experimental setup. a** Receiver architecture block diagram. **b** Fabricated die microphotograph (HF high frequency, BB baseband, LO local-oscillator, ED envelope detector). **c** Setup schematic for the receiver measurements (AWG arbitrary waveform generator). **d** Test equipment–lab test bench.

which the bit-error-rate (BER) is below 1e-3 as illustrated in Fig. 2b. In the measurement setup, the noise power level from the modulated input signal generator is fixed while only the input signal power is varied and the BER is recorded. Hence, by providing the receiver with different RF signal power levels, the analysis of the receiver performance at different input signal-to-noise ratios (SNRs) can be measured and has been presented. Measurements are performed for 5 Mbps and 1 Mbps data rates in normal mode and low-power mode, respectively.

Figure 2b shows that the proposed receiver has −62 dBm sensitivity at 5 Mbps and −67 dBm at 1 Mbps in normal mode and low-power mode, respectively, without any external passive RF voltage gain at the input. Based on the logarithmic relation between receiver sensitivity and receiver detector bandwidth[15], the sensitivity could be improved to around −77 dBm for 100 kbps data rate, which is comparable performance to the state-of-the-art works shown in Table 2. The trend could be observed in other receiver products, such as C1101 module[29]. Figure 2c illustrates the covered range of frequencies in which the proposed receiver can support higher data rates in normal and low-power modes. The transient waveforms of the receiver's input and output along with the test conditions are illustrated in Fig. 2d, e. The measured receiver latency is ~295 ns, which is highly competitive to the recently published works. The measured latency transient waveform is illustrated in Fig. 2f. The summary of the proposed receiver performance and comparison against the state-of-the-art RF-OOK receivers is recorded in Table 2. To have a fair comparison against the published state-of-the-art

receivers, energy efficiency is considered as the figure of merit and is defined as

$$\text{Energy efficiency}\,(\text{nJ per bit}) = \frac{\text{Power consumption}\,(\mu W)}{\text{Data rate}\,(\text{kbps})} \quad (1)$$

The energy efficiency of the receiver is better than most of the existing state-of-the-art OOK receivers from Table 2. The proposed receiver can handle a maximum data rate of 5 Mbps at very low power dissipation. The flexibility of normal and low-power modes offered by this design is highly suitable for applications that require uncompromised performance across a wide range of data throughput.

## Conclusion

This work reports the first highest data rate OOK receiver in the 2.4 GHz ISM band to date, to the author's knowledge. The proposed ultra-low power receiver is designed completely on-chip in CMOS 40 nm process, which can achieve a sensitivity of −62 dBm at the highest data rate of 5 Mbps consuming only 178 μW power. The desired target to support such higher data rates and very low power consumption, and small area is achieved with the help of the on-chip IMN, a unique pseudo-differential mixer topology, temperature-resilient VCRO without the need for external components, and a wideband envelope detector. The proposed receiver demonstrates one of the additional low-power modes that can achieve an improved sensitivity of −67 dBm at the maximum supported data rate of 1Mbps consuming only 119 μW power. In the low-power modes, the proposed receiver consumes 114.2 μW and 109.2 μW for 200 kbps and 100 kbps data rates, respectively. This provides design flexibility between higher data rates and improved RF sensitivity as required by the application. The sensitivity versus data rate trade-off could be explored further by customizing the bandwidth of the receiver system depending on the desired data rate. The complete receiver design, including the IO pads, occupies an area of 0.4mm². Compared to the previously published state-of-the-art OOK receivers, the proposed work achieves better performance in terms of support to at least 10 times higher

## Table 1 | Details of receiver modes of operation

| | Normal mode | | Low-power mode | |
|---|---|---|---|---|
| | $V_{DD}$ (V) | $I_{DC}$ (μA) | $V_{DD}$ (V) | $I_{DC}$ (μA) |
| Mixer and IF amplifier | 1 | 39.2 | 1 | 39.2 |
| VCRO | 0.9 | 74 | 0.9 | 74 |
| Envelope detector, comparator, and baseband amplifier | 1 | 72.3 | 0.7 | 19.1 |

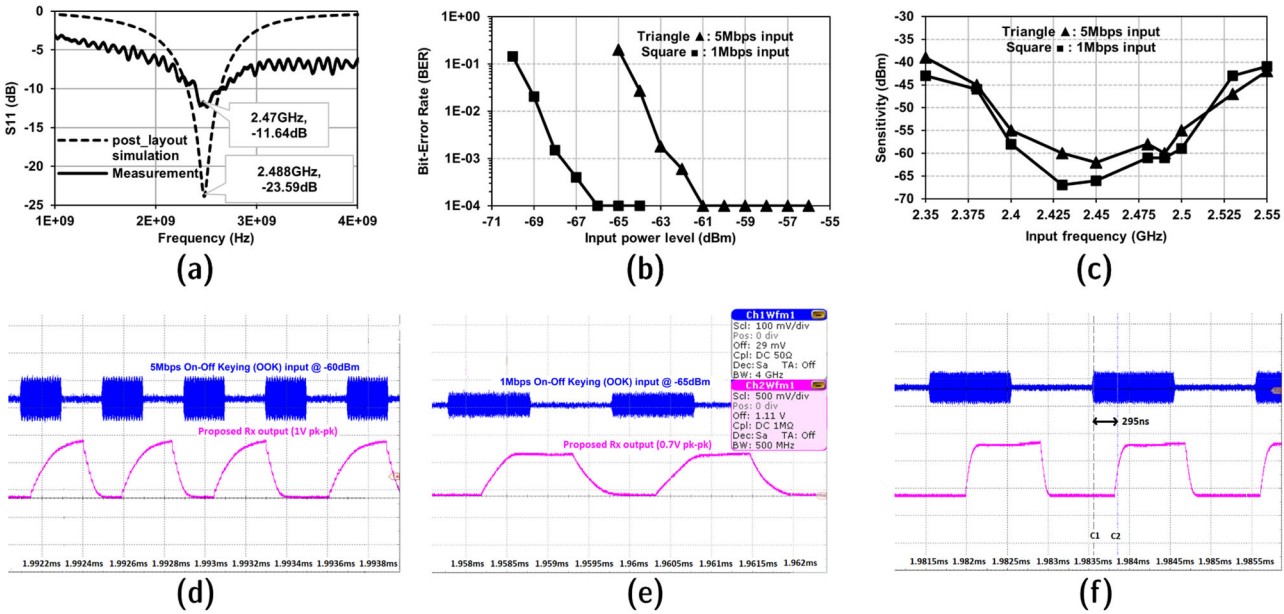

**Fig. 2 | Proposed receiver design performance. a** Simulated and measured data of input matching $|S_{11}|$. **b** Measured BER vs input signal power level. **c** Measured BER vs input signal frequency. **d** 5Mbps OOK decoding in normal mode. **e** 1 Mbps OOK decoding in low-power mode. **f** Latency measurement of the proposed receiver.

**Table 2 | State-of-the-art OOK receiver performance comparison with the proposed receiver**

| | [7] JSSC'09 | [8] ISSCC'10 | [18] JSSC'18 | [19] JSSC'19 | [22] JSSC'21 | [23] JSSC'21 | This work |
|---|---|---|---|---|---|---|---|
| Technology [nm] | 90 | 90 | 14 | 45 | 65 | 65 | 40 |
| Architecture | Mixer-first Uncertain-IF | Detector-first | Mixer-first | Low-IF | Mixer-first, N-path filtering | Mixer-based Zero second IF | Mixer first Low-IF |
| External components | BAW resonator | Inductors, Clocks | 32 kHz RTC | 2 MHz reference | FBAR, XO, Input matching | Input matching | None |
| LO generation | Unlocked RO | -- | Ring with FLL | LC-VCO | RO + PLL | RO + PLL + Frequency Tripler | Unlocked RO |
| Modulation | OOK | OOK | OOK | FSK | OOK | OOK | OOK |
| Power supply [V] | 0.5 | 0.5 | 0.95 | 1/0.9 | 0.5 | 0.5 | 1/0.9    1/0.9/0.7 |
| Frequency [GHz] | 2 | 2.4 | 2.4 | 2.4 | 2.4 | 2.4 | 2.4 |
| Power [µW] | 52 | 51 | 95 | 1200 | 220 | 352 | 178      119 |
| Latency [µs] | N.R. | N.R. | 6[a] | N.R. | 200[a] | 1470[a] | 0.295[b] |
| Data rate [ kbps] | 100 | 100 | 62.5 | 250 | 4 | 5 | 5000    1000 |
| Sensitivity @ 1e-3 BER [ dBm] | −72 | −64 | −72 | −82.2 | −85 | −92 | −62      −67 |
| Area [mm²] | 0.1* | 0.36** | 0.19** | 1** | 1.3** | 0.6** | 0.4** |
| Energy efficiency [nJ per bit] | 0.52 | 0.51 | 1.52 | 4.8 | 55 | 70.4 | 0.0356   0.119 |

*Core area.
**Area including IO pads.
N.R. means not reported. [a]Wake-up latency. [b]Receiver latency.

data rate, low power consumption, and at least 5 times better energy efficiency, as highlighted in Table 2.

## Methods/implementation
### Proposed mixer-first low-IF receiver
This section describes the presented receiver architecture along with the circuit-level topologies of the incorporated design blocks in the receiver. The proposed ultra-low power completely on-chip mixer-first receiver architecture is illustrated in Fig. 1a. This receiver design is capable of processing and demodulating the OOK modulated signals in the 2.4 GHz industrial, scientific, and medical (ISM) band. The LC-type IMN inherent passive gain characteristics[29] aid to amplify the incoming

RF signal at the receiver input. After passive amplification using an LC network, the RF signal is fed to a differential noise-canceling active mixer. The noise-canceling technique is incorporated in the mixer since the receiver noise performance depends on the first block in the architecture. In the mixer, the received RF signal is frequency down-converted with an internally generated LO signal at a slightly lower frequency compared to the received signal frequency. The stable LO signal is generated using a VCRO, and the frequency is resilient to temperature variations. The mixer output is at a low IF and is amplified using an IF amplifier. The output of the IF amplifier contains the message signal in the envelope which is detected by the envelope detector. The envelope detector's output is compared with an internally

generated reference signal for decision inference of data bits as '1' or '0'. The message signal with full swing between the rails is obtained by amplifying the comparator's output using the baseband amplifier. The ground connection of high-frequency blocks (mixer, IF amplifier, and local oscillator) is different from the ground connection of baseband blocks (envelope detector, comparator, and baseband amplifier) to avoid undesired oscillations that occur through the substrate leakage currents. Both the grounds are maintained at 0 V. These substrate leakage currents are due to the rapid amplification of the baseband amplifier in the receiver chain which delivers a rail-to-rail output signal. The effect of leakage currents is predominantly seen in system-on-chip (SoC) and system-in-package (SiP) solutions due to the presence of bond wire inductors. The bond wire inductors and the substrate capacitance along with other parasitics form a positive feedback loop including the active circuit in the receiver chain and produce spurious components. These random spikes eventually affect the signal transitioning path through the same substrate and affect the LO signal frequency and stability. Thus, the issue of random spikes is observed, and the proposed solution of separating ground for high-frequency and baseband blocks is verified through simulations. The design blocks integrated into the proposed receiver are described below in detail.

### Input matching network and inverter-based pseudo-differential active mixer.
The input matching network in the proposed receiver is designed using an LC network. The inductor and metal-oxide-metal capacitor are taken from the TSMC library with a decent quality factor. The value of the inductor chosen is ~27 nH, and the value of the capacitor chosen is ~150 fF. The quality factor of the chosen inductor is ~6, which is low, but since the motivation of this work is to obtain a completely on-chip receiver with low power consumption and comparable performance, it is considered the best choice. The IMN is designed on-chip to achieve a compact receiver.

The mixer topology used in this receiver design is unique and has various advantages, which are discussed in this section. The simplified schematic of the proposed active mixer is shown in Fig. 3a. In the proposed

mixer, an NMOS transistor and a PMOS transistor to feed the RF signal and LO signal, respectively, are used. The output of the circuit is analyzed at the common drain of these transistors. The circuit functionality is demonstrated by using a square-wave LO signal and a sinusoidal RF input. In the actual scenario, the input RF signal is OOK modulated, and hence, the signal component is essentially a sine wave in transient mode when the message signal bit is 1. The PMOS transistor is either ON or OFF, depending on the 50% duty-cycled LO signal. The NMOS transistor is internally biased, and thus, the transconductance of the NMOS transistor will also follow a square-wave function due to the DC path consisting of the LO-driven PMOS transistor. So, the transconductance of the NMOS transistor is given by the Fourier expansion of a square wave with amplitude $g_m$ in Eqs. (2) and (3).

$$g_{mn1} = \frac{g_m}{2} + \sum_n \frac{2g_m}{n\pi} \sin\left(\frac{n\pi}{2}\right) \sin(n\omega_{LO}t) \tag{2}$$

$$g_{mn2} = \frac{g_m}{2} - \sum_n \frac{2g_m}{n\pi} \sin\left(\frac{n\pi}{2}\right) \sin(n\omega_{LO}t) \tag{3}$$

From Eqs. (2) and (3), the expressions for $v_{outp}$ and $v_{outn}$ can be deduced as shown in (4) and (5).

$$v_{out,p} = g_{mn1}v_{RF}Z_L = A.m(t)\left[\frac{g_m}{2}\sin(\omega_{RF}t) + \frac{g_m}{\pi}\cos(\omega_{IF}t) - \frac{g_m}{\pi}\cos((\omega_{RF}+\omega_{LO})t) + \right]Z_L \tag{4}$$

$$v_{out,n} = g_{mn2}v_{RF}Z_L = A.m(t)\left[\frac{g_m}{2}\sin(\omega_{RF}t) - \frac{g_m}{\pi}\cos(\omega_{IF}t) + \frac{g_m}{\pi}\cos((\omega_{RF}+\omega_{LO})t) - \right]Z_L \tag{5}$$

The generated differential output in Eqs. (4) and (5) can be capacitively coupled to the differential IF amplifier (5T-OTA) to amplify the desired IF signal and eliminate the RF signal component. The higher-order mixing components are also attenuated abiding by the bandpass filter characteristics

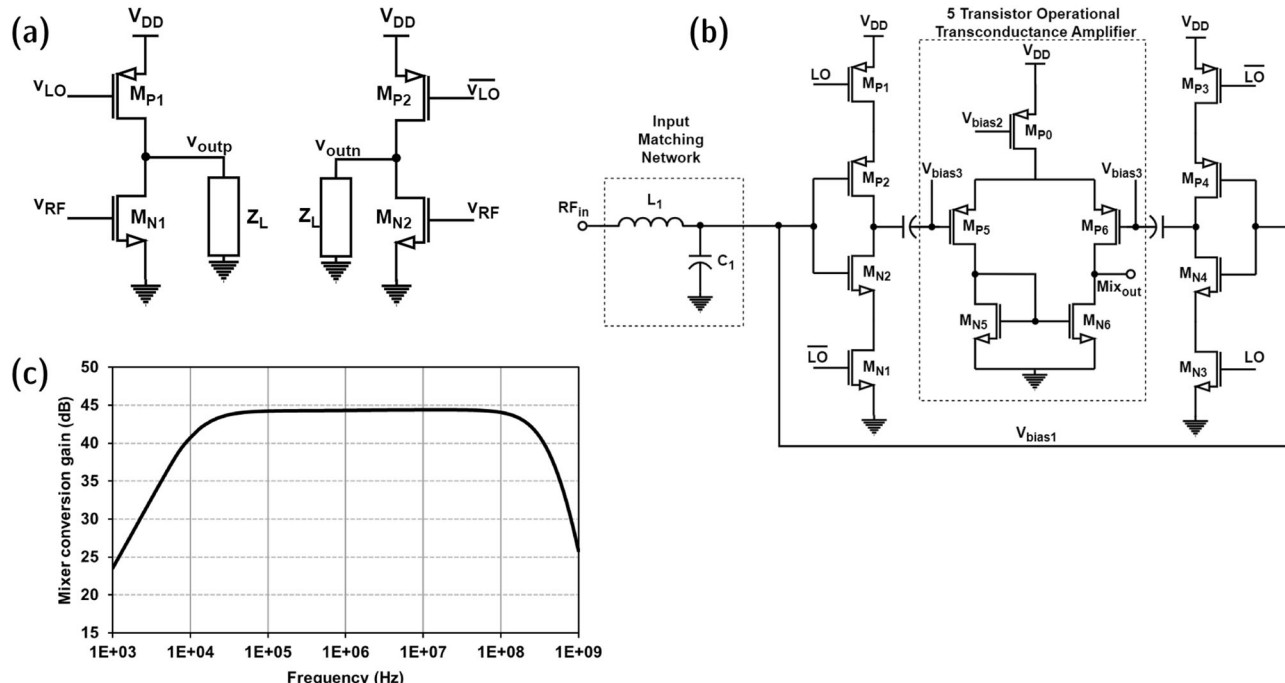

**Fig. 3 | Proposed pseudo-differential active mixer. a** Simplified schematic (LO local-oscillator, RF radio-frequency). **b** Full schematic in detail (Mix$_{out}$ mixer output). **c** Simulated voltage conversion gain.

of the 5T-OTA. For the proposed pseudo-differential active mixer design, the $g_m$ term in Eqs. (2–5) is effectively the sum of transconductances of NMOS ($M_{N2}$, $M_{N4}$) and PMOS ($M_{P2}$, $M_{P4}$) transistors shown in Fig. 3b.

The proposed pseudo-differential active mixer topology is shown in Fig. 3b. The inverter-based active mixer consists of two NMOS and two PMOS transistors. The PMOS transistor near the supply voltage node and the NMOS transistor near the ground node are driven by complimentary LO signals, as shown in the schematic. The gates of NMOS and PMOS transistors between these LO-driven transistors are connected. This inverter structure is driven by the receiver RF input signal after passive amplification from the matching network. The aspect size ratio of PMOS to NMOS transistors is maintained to be 3:1 so that maximum swing can be achieved at the inverter stage output. To improve the noise performance of the mixer, an internally differential topology is proposed where the input of the mixer is single-ended, and the output of the mixer is also single-ended, but internally, the two branches of the mixer generate a differential signal which is fed to a differential-to-single-ended 5T-OTA (D2S 5T-OTA). This technique helps to cancel out the noise generated by the mixer down-conversion while amplifying the desired signal component. The mixer performance is analyzed at the post-layout level using AC simulations. The mixer integrated with the 5T-OTA provides a voltage gain of 44 dB over a wide range of frequencies from 1 MHz to 125 MHz. The mixer noise figure is around 15.3 dB at a power dissipation of 24 µW. The RF-IF and LO-IF isolation is very large as the RF input alone is matched to 50 Ω. The LO port and IF port are AC-coupled and low-pass filtered, respectively, thus attenuating the RF and LO signals at the mixer output and further attenuated by the IF amplifier. The simulation plot of AC voltage gain for the mixer is illustrated in Fig. 3c. The LO signal is provided by a VCRO, and the circuit details are discussed below.

**Voltage-controlled ring oscillator.** The proposed receiver architecture requires a stable frequency signal for the down-conversion of the received RF signal. Most of the ring oscillators generate either sinusoid or rail-to-rail signals depending on the design requirements. In this work, a unique technique is proposed that will minimize the deviation of the frequency of the generated LO signal with respect to temperature variations. The proposed VCRO is illustrated in Fig. 4a. With an increase in temperature, the ring oscillator frequency can either increase or decrease depending on two factors. One factor is the transistor threshold voltage, and the other is transistor mobility. In lower technology nodes, the factor of the threshold voltage is predominant compared to the mobility factor.

In Fig. 4a, the VCRO has 3 inverter stages cascaded, and the output of the third inverter stage is connected to the input of the first inverter stage. The source of NMOS transistors is connected to the drain of the current-controlling NMOS transistor ($M_{N0}$), and the source of PMOS transistors is connected to the drain of the current-controlling PMOS transistor ($M_{P0}$), as shown in Fig. 4a. The current-controlling transistors require gate voltage to appropriately neutralize the effect of temperature on the transistors in the inverter stages. The transistors $M_{N0}$ and $M_{P0}$ are in saturation region where the transistor resistance increases with an increase in temperature. This behavior is used to complement the effect of temperature on the ring oscillator frequency. The $M_{P0}$ and $M_{N0}$ transistors' output resistance when operated in the saturation region is shown in Fig. 4b using simulated results. The oscillator is designed to generate a square-wave signal using inverter-based buffer amplifiers with a frequency of 2.39 GHz for a voltage supply of 0.9 V. The VCRO resilience to the variations in temperature is illustrated in Fig. 4c through simulation results. Thus, for the input RF signal frequency between 2.4 and 2.49 GHz, the down-converted IF signal frequency lies between 10 MHz and 100 MHz, which is amplified using the IF amplifier discussed below. The deviation of VCRO signal frequency w.r.t process variations is taken into consideration, and thus, the frequency tuning can be achieved through the supply voltage of the VCRO block as illustrated in Fig. 4d. The proposed VCRO topology can incorporate the input-coupling technique[30] to enhance the phase noise performance comparable to LC-based VCOs. In the proposed solution as a heterodyne OOK receiver, the phase noise of the VCRO is not critical[15] since the IF frequency can be

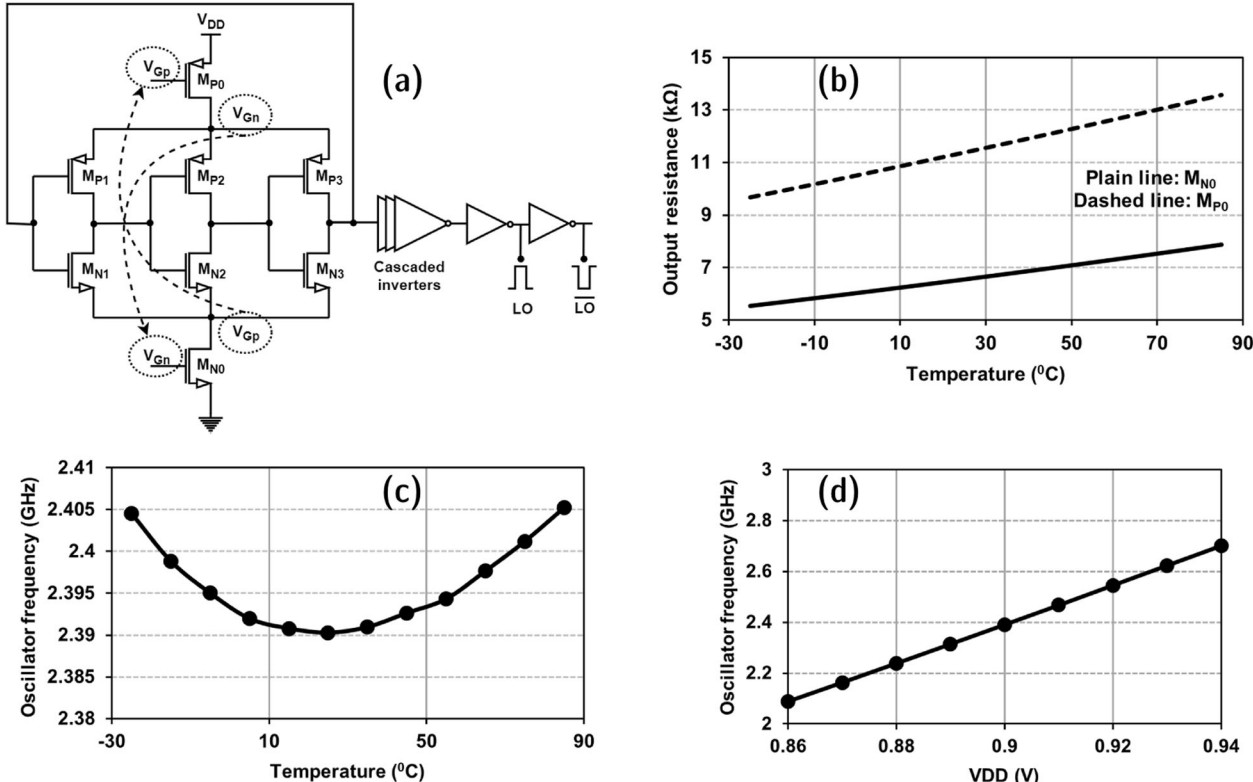

**Fig. 4 | Proposed voltage-controlled ring oscillator. a** Schematic (LO: local-oscillator). **b** Simulated variation of output resistance of NMOS and PMOS transistors in saturation region w.r.t temperature. **c** Simulated frequency of VCRO output signal with respect to temperature. **d** Simulated VCRO frequency tuning with respect to $V_{DD}$.

uncertain and lie within the supported IF bandwidth for decoding amplitude-modulated signals. Thus, the proposed VCRO topology is chosen to provide a stable LO signal with ultra-low power consumption[31–37].

**IF amplifier.** The IF amplifier in this receiver is designed using self-biased inverter stages cascaded together, as shown in Fig. 5a. The IF amplifier plays a major role in amplifying the IF signal at the mixer output. The mixer output contains a considerable power level of LO signal and its harmonics due to the chosen passive internally differential mixer topology. Thus, the IF amplifier should have bandpass filter characteristics along with the desired gain. The IF amplifier input is AC coupled with the mixer output to prevent the mixer output DC operating point from driving the IF amplifier unit cell into saturation. The designed IF amplifier consists of 3 unit cells cascaded to achieve maximum gain. The cascading is achieved through AC coupling to prevent signal saturation along the path. The IF amplifier unit cell is a self-biased inverter. The self-biasing is obtained through a large resistive connection between the input and the output of the inverter. The detailed schematic is shown in Fig. 5a.

To achieve a temperature and voltage-resilient IF amplifier design, the input-to-output resistor symbolically illustrated in Fig. 5b is designed using an active device and a gate control voltage. The gate control voltage $V_g$ is generated using the internal bias circuit shown in Fig. 5c to minimize the deviation of amplifier performance with variations in supply voltage and temperature. The bandpass filter characteristics and the post-layout voltage gain of the IF amplifier are depicted in Fig. 5d. The simulated gain of the IF amplifier is around 33 dB with a 3 dB bandwidth of 125 MHz with a peak at 25 MHz. The empirical expression for the gain of the single stage of the proposed IF amplifier is provided in Eq. (6) as a function of transconductances of NMOS and PMOS transistors in each stage and the effective resistance of $M_{N0}$ and $M_{P0}$. This is a simplified equation under the assumption that the output resistance of transistors ($r_{0n1}$, $r_{0p1}$) is large compared to the effective resistance ($R_{eff}$).

$$A_{v,\text{single–stage}} = -\left(g_{mn1} + g_{mp1}\right)R_{eff} + 1 \qquad (6)$$

The variation of $R_{eff}$ is complementary to the variation of $g_{mn1} + g_{mp1}$ w.r.t voltage and temperature variations. This is achieved using the designed

bias circuit using transistor $M_{P4}$ and diode $D_1$ in Fig. 5c. With an increase in supply voltage, the transconductance increases, and with an increase in temperature, the transconductance decreases. The exact opposite trend is achieved for the effective resistance to achieve the robustness of IF amplifier gain characteristics. The variation of effective resistance and corresponding gate-bias voltage $V_g$ with respect to voltage and temperature variations are shown in Fig. 6a, b, which is obtained through simulations. The simulated IF amplifier gain response and its robustness against temperature and voltage variations are illustrated in Fig. 6c, d. The envelope of the amplified IF signal is detected using the envelope detector circuit discussed below.

**Envelope detector.** The amplified IF signal is passed through the envelope detector to detect the data present in the IF signal. Since the received signal is OOK modulated, the down-converted IF signal contains the IF carrier modulated by the data signal. The envelope detector proposed in this receiver is adapted from a conventional source-follower topology. Since the receiver architecture is chosen to be a low-IF design, the envelope detector is AC coupled to the IF amplifier to minimize the 1/f noise. The ED circuit consists of two phases namely envelope detection and envelope amplification. The IF signal is fed to the gate of $M_{N2}$, and the detected envelope is amplified using the single IF-amplifier stage, as shown in Fig. 7a. The bias voltage to $M_{N1}$ and $M_{N2}$ is generated internally to minimize the process-voltage-temperature variations. Transistor $M_{N2}$ is biased in the saturation region to realize the square-law function, which decodes the envelope of an OOK-modulated IF signal. The transistor sizes are designed accordingly to support higher data rate signals which require a relatively larger bandwidth. The envelope is amplified to ease the requirement of the comparator stage with an accurate reference voltage. If the envelope has a sufficient amplitude, then the comparator reference voltage need not be so precise, which is directly related to the resolution of the comparator. This is suitable for low-power applications.

**Comparator.** The comparator design plays an important role in RF receivers whose architectures are based on envelope detection. The envelope detector stage output is single-ended, and thus, the comparator requires a reference voltage to compare the input signal and amplify the envelope with the appropriate duty cycle. To achieve the desired

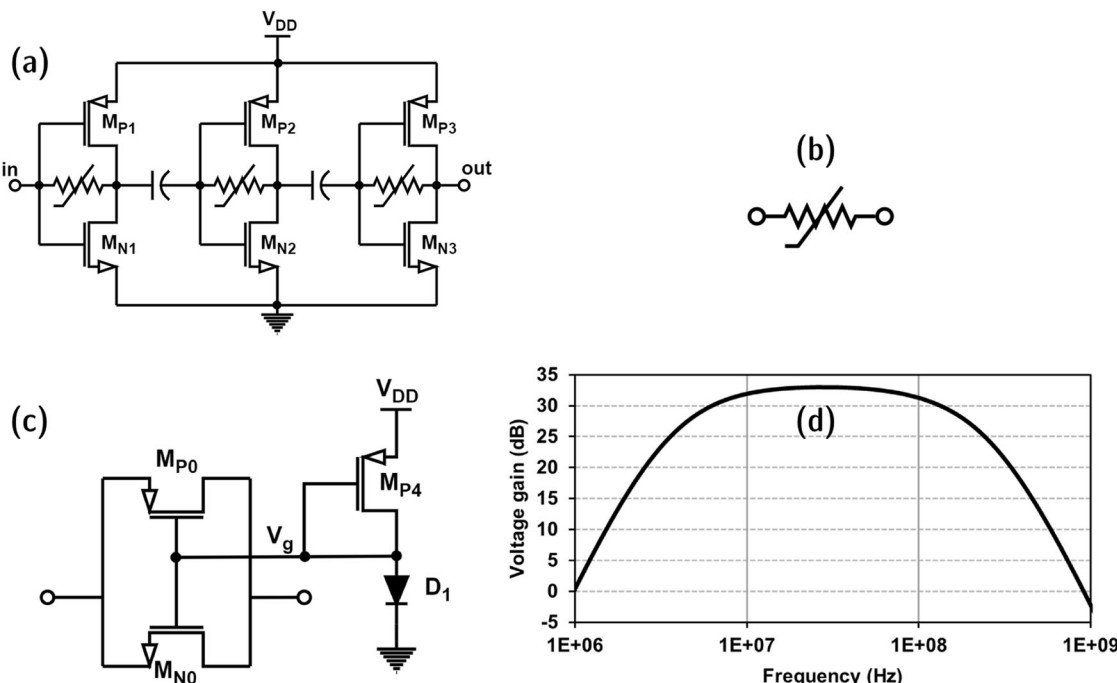

**Fig. 5 | Proposed IF amplifier design block. a** Schematic of the IF amplifier with bandpass filter characteristics. **b** Variable resistor symbol. **c** Equivalent circuit schematic of the variable resistor. **d** Simulated bandpass voltage gain characteristics of IF amplifier.

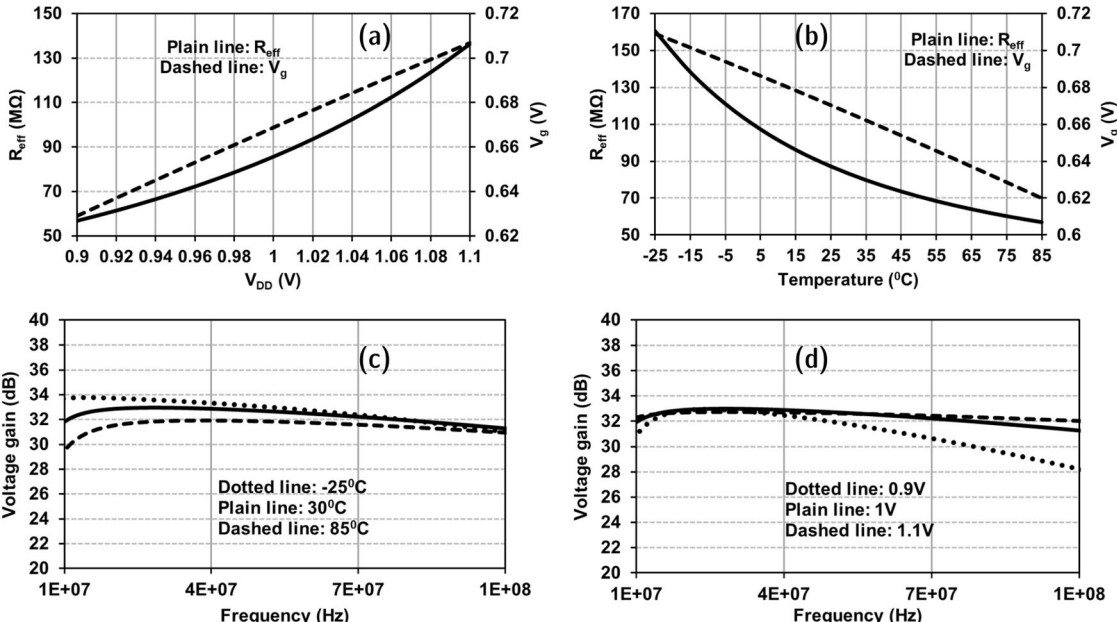

**Fig. 6 | Performance analysis of IF amplifier. a** Effect of supply voltage on gate-bias voltage ($V_g$) and combined effective resistance ($R_{eff}$) of $M_{N0}$ and $M_{P0}$. **b** Effect of temperature on gate-bias voltage ($V_g$) and combined effective resistance ($R_{eff}$) of $M_{N0}$ and $M_{P0}$. **c** Voltage gains robustness with temperature variation. **d** Voltage gain robustness with supply voltage variation.

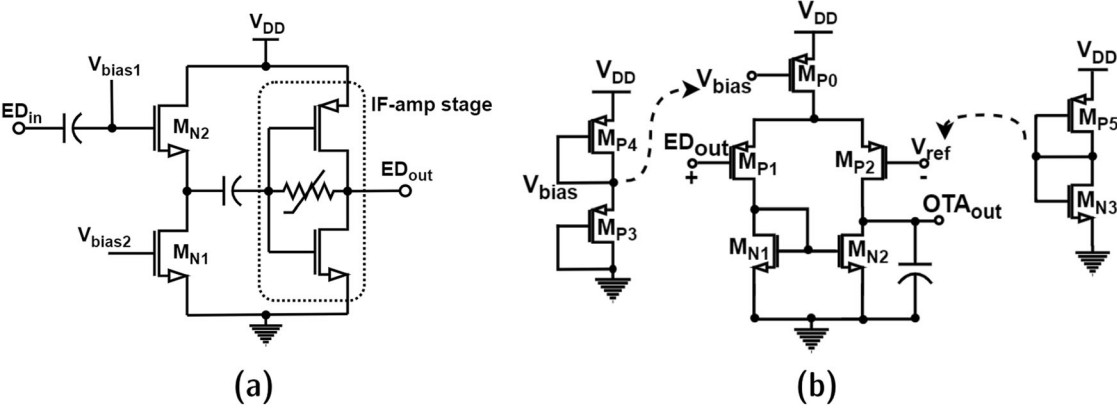

**Fig. 7 | Baseband blocks—envelope detector and comparator. a** Schematic of the implemented source-follower envelope detector (ED envelope detector, IF-amp intermediate frequency amplifier). **b** Schematic of the implemented comparator block (OTA: operational transconductance amplifier).

**Fig. 8 |** Baseband amplifier—circuit schematic of the proposed baseband amplifier (OTA operational transconductance amplifier, BB baseband).

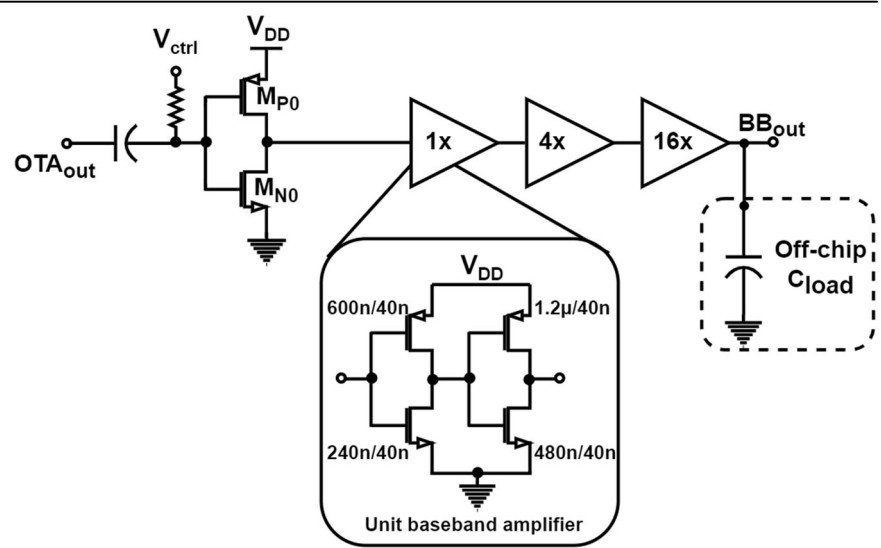

performance from the comparator at smaller power dissipation, the comparator circuit topology is chosen to be a differential 5T-OTA. Two inputs of the comparator are provided by the envelope detector output and an internally generated DC bias voltage signal. The envelope detector's output stage replica is used to generate the DC bias as a reference signal for the comparator to minimize process mismatches. The comparator input is DC-coupled to the envelope detector output which is also an added advantage since the requirement of an additional biasing circuit is alleviated. The complete circuit schematic of the comparator is shown in Fig. 7b. Transistors with a lower threshold voltage (low-$V_T$) are used to optimize the comparator gain and support higher data rates, at very low power consumption. The comparator's output is further amplified using the baseband amplifier to achieve rail-to-rail swing of the data signal.

**Baseband amplifier**. For low-power consumption, a simple topology is chosen for the baseband amplifier, which is an inverter-based amplifier. The comparator's output is AC coupled to the baseband amplifier, and thus, it requires an appropriate bias voltage to achieve the correct duty cycle for the data bits at the output. The bias voltage is labeled as $V_{ctrl}$ as shown in Fig. 8. The transistor size of $M_{N0}$ and $M_{P0}$ are 120n/40n and 300n/40n, respectively. This bias voltage is externally controlled during the measurements to maintain a 50% duty cycle at the baseband amplifier output. The inverter-based amplifiers are cascaded in six stages with increasing transistor sizes to be able to drive the large load capacitance. The baseband amplifier is designed to be capable of driving a large load capacitance of 15 pF from measurement equipment at higher data rates.

## Data availability
The authors declare that the data supporting the findings of this study are available within the paper.

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

## Author contributions

Conceptualization: A.B., B.K.T. circuit design: A.B., B.K.T. data collection: A.B. data analysis: A.B. funding acquisition: K.S.Y. writing–original draft: A.B. writing–review and editing: A.B., B.K.T., K.S.Y.

## Competing interests

The authors declare no competing interests.
