## [Peer Review File · Communications Engineering]

Reviewer #1

This work proposes an on-chip solution for a low-power OOK receiver front end that achieves the best energy efficiency at a high OOK data rate compared to other works. The authors presented block-level architectures of the on-chip receiver in detail. This paper is overall well-written with potential improvements. Here are a few questions or concerns for the authors.

The authors should provide examples of what kind of wake-up energy-harvesting applications can benefit from a high data rate to strengthen the advantage of this design.

This work achieves the highest data rate and energy efficiency. However, the sensitivity is at a moderate level compared to prior works. Fig.1.(d) and (e) show specs related to input power and sensitivity at high rates of 5Mbps and 1Mbps. The authors could discuss if these numbers change at lower data rates like 200 kbps and 100 kbps mentioned in the conclusion section.

The measurement was done in the lab environment. Analysis related to receiving signals with different signal-to-noise ratios (SNRs) was not conducted. Showing SNR results can consolidate the functionality of the proposed chip or system.

Reviewer #2

The paper claims the development of the first highest data rate On-Off Keying (OOK) receiver in the 2.4 GHz ISM band, achieving a sensitivity of -62 dBm at 5 Mbps while consuming only 178 μ W. The proposed receiver offers design flexibility between higher data rates and improved RF sensitivity, outperforming previous state-of-the-art OOK receivers in terms of data rate, power consumption, and energy efficiency. However, the reviewer has major concerns about the quality of the work.

The paper's overall organization falls short of effectively presenting its work in a structured manner. The abstract appears more akin to the beginning of an introduction section rather than serving its intended purpose. The introduction, while containing a brief and inadequate literature review, could benefit from a more extensive exploration to establish a meaningful connection between the work's requirements and the potential novelty of its proposed solution.

Given that the presented design is based on die package fabrication, it would be more fitting to draw comparisons and incorporate prior literature specifically related to this method.

Table II should include a comparison of Bit Error Rate (BER) for each design, and the design itself should be benchmarked against others fabricated using die package methods.

Some pertinent comments on the results are as follows. For instance, in VCRO section, crucial information such as phase noise value, mixer isolation, and overall design power efficiency are prominently absent. The fabrication process details, particularly those related to transistor-based circuits, necessitate a more comprehensive explanation.

Furthermore, the paper could benefit significantly from the inclusion of a software-based layout graphic to enhance the clarity of the presentation.

Rebuttal

The authors would like to thank the reviewers for their time and effort in providing valuable comments to improve our work presentation.

Reviewer #1

This work proposes an on-chip solution for a low-power OOK receiver front end that achieves the best energy efficiency at a high OOK data rate compared to other works. The authors presented block-level architectures of the on-chip receiver in detail. This paper is overall well-written with potential improvements. Here are a few questions or concerns for the authors.

The authors should provide examples of what kind of wake-up energy-harvesting applications can benefit from a high data rate to strengthen the advantage of this design.

This work achieves the highest data rate and energy efficiency. However, the sensitivity is at a moderate level compared to prior works. Fig.1.(d) and (e) show specs related to input power and sensitivity at high rates of 5Mbps and 1Mbps. The authors could discuss if these numbers change at lower data rates like 200 kbps and 100 kbps mentioned in the conclusion section.

The measurement was done in the lab environment. Analysis related to receiving signals with different signal-to-noise ratios (SNRs) was not conducted. Showing SNR results can consolidate the functionality of the proposed chip or system.

(1) The authors should provide examples of what kind of wake-up energy-harvesting applications can benefit from a high data rate to strengthen the advantage of this design.

[Reply]: Thank you for the valuable comment. We have added a few sentences on the specific applications that could benefit from high data rate communication such as environmental monitoring, industrial IoT, smart agriculture.

[Modifications in the revised manuscript]:

(In introduction, pg. 2, line. 59, text included) “Wireless sensor networks (WSNs) can benefit from high data rates to allow real-time decoding for environmental monitoring, industrial IoT, and smart agriculture. Healthcare monitoring devices benefit from high data rate in decoding vital signs through remote monitoring in real-time. Other applications include smart home automation, asset tracking, inventory management, and cognitive cameras with high volume data processing.”

(2) This work achieves the highest data rate and energy efficiency. However, the sensitivity is at a moderate level compared to prior works. Fig.1.(d) and (e) show specs related to input power and sensitivity at high rates of 5Mbps and 1Mbps. The authors could discuss if these numbers change at lower data rates like 200 kbps and 100 kbps mentioned in the conclusion section.

[Reply]: Thank you for the valuable comment. Though the sensitivity is at moderate level, the corresponding data rate is higher than the prior works. The receiver

sensitivity varies logarithmically on the receiver detector bandwidth in the proposed architecture. To support high data rates, it is essential to provide sufficient bandwidth equivalent to $\text{data rate}/2$ (in hertz) at the envelope detector stage. However, a wider bandwidth at the envelope detector leads to poor sensitivity based on the relation between receiver sensitivity and detector bandwidth is logarithmic as mentioned in [7] as follows.

RX Sensitivity = $-174 + 10 \cdot \log(\text{Detector bandwidth}) + \text{Total receiver noise figure} + \text{minimum required baseband SNR}$. (from [7])

$$P_{Sens} = -174 \text{dBm} + 10 \log BW_{det} + NF_{tot} + SNR_{min}$$

The receiver sensitivity could be improved for lower data rates which require lesser detector bandwidth. For 100kbps data rate, the receiver sensitivity could improve to -77dBm (10dB more) from -67dBm at 1Mbps data rate.

[Modifications in the revised manuscript]:

1. (In pg. 5, line. 183, text included) “Based on the logarithmic relation between receiver sensitivity and receiver detector bandwidth^[13], the sensitivity could be improved to around -77dBm for 100kbps data rate which is comparable performance to the state-of-the-art works shown in Table II. The trend could be observed in other receiver products such as C1101 module^[30].”
2. (In references, reference added) “[30] Texas Instruments. CC1101 Low-Power Sub-1 GHz RF Transceiver Datasheet (2013).”
3. (In conclusion, pg. 6, line. 217, text included) “The sensitivity versus data rate trade-off could be explored further by customizing the bandwidth of the receiver system depending on the desired data rate.”

(3) *The measurement was done in the lab environment. Analysis related to receiving signals with different signal-to-noise ratios (SNRs) was not conducted. Showing SNR results can consolidate the functionality of the proposed chip or system.*

[Reply]: Thank you for the valuable comment. This is a noteworthy approach to support and compliment the receiver functionality. However, the suggested experiment could not be performed due to the limitations in the available waveform generators from the laboratory. In the measurement setup, the noise power level from the modulated input signal generator is fixed while only the input signal power is varied. By analysing the receiver performance at different RF signal power levels, we are indirectly analysing the receiver with different SNRs since the noise power is fixed.

[Modification in the revised manuscript]:

1. (In pg. 5, line. 167, text included) “In the measurement setup, the noise power level from the modulated input signal generator is fixed while only the input signal power is varied and the BER is recorded. Hence, by providing the receiver with different RF signal power levels, the analysis of the receiver performance at different input signal-to-noise ratios (SNRs) can be measured and has been presented.”

Reviewer #2

The paper claims the development of the first highest data rate On-Off Keying (OOK) receiver in the 2.4 GHz ISM band, achieving a sensitivity of -62 dBm at 5 Mbps while consuming only 178 μ W. The proposed receiver offers design flexibility between higher data rates and improved RF sensitivity, outperforming previous state-of-the-art OOK receivers in terms of data rate, power consumption, and energy efficiency. However, the reviewer has major concerns about the quality of the work.

The paper's overall organization falls short of effectively presenting its work in a structured manner. The abstract appears more akin to the beginning of an introduction section rather than serving its intended purpose. The introduction, while containing a brief and inadequate literature review, could benefit from a more extensive exploration to establish a meaningful connection between the work's requirements and the potential novelty of its proposed solution.

Given that the presented design is based on die package fabrication, it would be more fitting to draw comparisons and incorporate prior literature specifically related to this method.

Table II should include a comparison of Bit Error Rate (BER) for each design, and the design itself should be benchmarked against others fabricated using die package methods.

Some pertinent comments on the results are as follows. For instance, in VCRO section, crucial information such as phase noise value, mixer isolation, and overall design power efficiency are prominently absent. The fabrication process details, particularly those related to transistor-based circuits, necessitate a more comprehensive explanation.

Furthermore, the paper could benefit significantly from the inclusion of a software-based layout graphic to enhance the clarity of the presentation.

(1) The paper's overall organization falls short of effectively presenting its work in a structured manner. The abstract appears more akin to the beginning of an introduction section rather than serving its intended purpose. The introduction, while containing a brief and inadequate literature review, could benefit from a more extensive exploration to establish a meaningful connection between the work's requirements and the potential novelty of its proposed solution. Given that the presented design is based on die package fabrication, it would be more fitting to draw comparisons and incorporate prior literature specifically related to this method.

[Reply]: Thank you for the constructive comment. As suggested, we have improved the structure of our work presentation abiding to the journal guidelines and the manuscript is written based on the journal's template.

Please refer here.

[https://www.nature.com/commseng/submit/submission-guidelines#:~:text=Article%20formatting%20guidelines%3A%20an%20abstract,items%20\(figures%2C%20tables\)](https://www.nature.com/commseng/submit/submission-guidelines#:~:text=Article%20formatting%20guidelines%3A%20an%20abstract,items%20(figures%2C%20tables)).

We made the following changes in the abstract, introduction and conclusion sections in the revised manuscript based on the suggestions. We developed the content flow by establishing the connection between the requirements and novelty based on

extensive literature review. We verified and validated that all the works incorporated in the literature study are die fabrication-based designs.

We modified the abstract providing a general implication of our results with a short description on the importance of this work.

[Modifications in the revised manuscript]:

1. (In abstract, text included and modified) ~~“Internet-of-things (IoT) devices has a huge demand for power sources. For sustainability, energy harvesting technologies are currently pursued as a potential power source for these IoT devices. In certain applications where continuous operation mode is required, these power sources get exhausted soon. Most of the power is dissipated during wireless communication of IoT devices. As a result, it is crucial to achieve low power solutions for high speed data communications. Here we show a fully integrated radio frequency (RF) receiver frontend with unique block level circuit design techniques to decode high data rate On-Off Keying (OOK) modulated signals in the 2.4 GHz Industrial, Scientific, and Medical (ISM) band. As a result, this work provides a compact design while avoiding bulky external components such as BAW resonators, crystal oscillators, etc. The proposed OOK receiver achieves a sensitivity of -62dBm/-67dBm while consuming 178 μ W/119 μ W power for a 5Mbps/1Mbps data rate. Despite the enormous potential of energy-efficient receivers for wireless sensor networks (WSN), the large power consumption or limited data rate support impedes its extensive applications. Here, we present an energy-efficient, ultra-low power, higher data rate supporting, completely on-chip radio frequency (RF) receiver frontend for On Off Keying (OOK) modulated signals in the 2.4GHz Industrial, Scientific, and Medical (ISM) band. This compact-sized receiver is achieved by implementing temperature-resilient oscillator, pseudo-differential mixer, and a wideband detector while avoiding bulky external components such as BAW resonators, crystal oscillators. Measurement results demonstrate that the proposed OOK receiver can decode low power level RF signals up to 5Mbps data rate while consuming only 178 μ W power. This work also demonstrates support for lower data rates at reduced power. Since the proposed receiver can in different power modes, it can be integrated in diverse applications including Internet-of-Things (IoT) devices and continuously monitoring biomedical/wearable implants.”~~
2. (In introduction pg. 2, line. 52, text included and modified) ~~“IoT-based applications demand higher data rates depending on the type of information shared through wireless technologies. To address the demand for higher throughput in wireless communications, the transceiver systems should be capable of handling the information data bits at a much higher speed. In IoT, the data communication at rapid pace is very important^[5]. This demand comes from the immense scope of applications that could be addressed and fulfilled through some unique techniques in the integrated circuit design^[6]. For example, the society sees many cases of fatality that could be prevented if only the cause was noticed a few seconds earlier. This often occurs in healthcare and other critical industries. Thus, it is important to establish a system which can detect errors within the least time after a trigger occurs^[7]. Translating the scenario to wireless communications, it is desired to decode information with minimum latency. Not only is the latency that is important, but also the amount of information which is represented by the data rate of the signal^[8]. Wireless sensor networks (WSNs) can benefit from high data rates to allow real-time~~

decoding for environmental monitoring, industrial IoT, smart agriculture, etc. Healthcare monitoring devices benefit from high data rate in decoding vital signs through remote monitoring in real-time^[9]. Other applications include smart home automation, asset tracking and inventory management, cognitive cameras with high volume data processing, etc. The constraints on power consumption and area comes from a wide set of applications in WSNs including biomedical implants, remote sensing, and surveillance^[10].”

3. (In introduction pg. 2, line. 66, text included and modified) ~~“Future wireless communications require high data rate supported radio frequency integrated circuits (RFICs) which are reliable and are better performing in terms of the desired parameters such as the power consumption, data throughput, link bandwidth, receiver’s sensitivity, and transmitter output power^[6]. Considering all these factors, it is essential to provide a sustainable and long-lasting robust solution for usability and reproducibility for mass production. In the case of a radio frequency (RF) receiver, the support for higher data rate and better sensitivity at very low power consumption is a critical challenge. These wireless network applications require highly integrated electronic solutions with minimum silicon area for reduced costs and very low power consumption for better battery life^[7]. To achieve these goals, the choice of receiver architecture plays a crucial role and predetermines certain important trade-offs that need to be met. Unique design techniques must be incorporated in order to achieve high data rate support and low latency at lower power consumption and small area. It is essential to support high data-rate communication to facilitate and aid a wide range of applications.”~~
4. (In introduction pg. 2, line. 77, text included) ~~“Some designs have presented a very good sensitivity at high-power consumption and decent data rates. Some designs have presented a good sensitivity at a very low power consumption but for very low data rates. Some other designs have presented a reasonable sensitivity at moderate power consumption and moderate data rates. This literature study suggests that a trade-off between sensitivity, power consumption, and data rate prevails in receivers.”~~
5. (In introduction pg. 3, line. 118, text included and modified) ~~“Prior to fabrication, the proposed receiver functionality is verified through simulations performed through electronic design automation (EDA) software tools. A low-power high data rate support mixer-first receiver is presented. The proposed receiver also incorporates a stable on-chip local oscillator (LO) signal generated from a modified voltage-controlled ring oscillator (VCRO) topology. This LO can be directly used in applications where the phase noise of the LO signal is not a major specification. There is also a provision to tune the frequency of the generated LO signal which is briefly detailed in the methods section. The miniature sized receiver is achieved by considering a compact layout design and on-chip input matching compromising the additional RF front-end gain through external resonators in the matching network which could improve the receiver sensitivity by at least 17 – 20dB^[13]. The proposed receiver is realized at a very low power consumption by considering a mixer-first receiver architecture as the LNA in the RF front-end could consume a significant amount of power. In the proposed mixer-first receiver, a unique inverter-based active mixer topology is designed that consumes only 24 μ W of power while providing an RF-to-DC conversion gain of more than 40dB including some passive RF voltage gain through the IMN. The proposed receiver also incorporates a stable~~

on-chip local-oscillator (LO) signal generated from a modified voltage-controlled ring oscillator (VCRO) topology. This LO can be directly used in applications where the phase noise of the LO signal is not a major specification. There is also a provision to tune the frequency of the generated LO signal which is briefly detailed in the methods section. The local oscillator generates a signal to down-convert the RF signal to an intermediate frequency (IF). A temperature resilient current-starved voltage-controlled ring-oscillator is designed which consumes very little power and aids to the overall power budget of the receiver. The high data rate support at low power is achieved by using a wideband envelope detector. Thanks to the pseudo-differential characteristics of the mixer comprising single-ended RF and differential LO which inherently generates a differential IF signal to the 5-transistor operational transconductance amplifier (5T-OTA) for additional gain. Thus, with the combination of an on-chip IMN, a pseudo-differential inverter-based active mixer, a temperature resilient current-starved ring-oscillator, and a wideband source-follower envelope detector, the proposed receiver is designed to occupy core die area of 0.122mm² to support 5Mbps/1Mbps data rate with 295ns latency at 178μW/119μW of power consumption achieving -62dBm/-67dBm sensitivity.”

6. (In conclusion, pg. 6, line. 208, text included) “The desired target to support such higher data rates as 5Mbps and very low power consumption of 178μW and small area of 0.122mm² are achieved with the help of the on-chip IMN, a unique pseudo-differential mixer topology, temperature-resilient VCRO without the need of external components, and a wideband envelope detector.”
7. (In references, reference added)
“[5] C. Perera, A. Zaslavsky, P. Christen and D. Georgakopoulos. Context aware computing for the internet of things: A survey. IEEE Communications Surveys & Tutorials 16, 414–454 (2013).
[6] Tang, J., Wang, Q., Tian, J. et al. Low power flexible monolayer MoS₂ integrated circuits. Nat Commun 14, 3633 (2023).
[7] Powell, D., Godfrey, A. Considerations for integrating wearables into the everyday healthcare practice. npj Digit. Med. 6, 70 (2023).
[8] Avellar, L., Stefano Filho, C., Delgado, G. et al. AI-enabled photonic smart garment for movement analysis. Sci Rep 12, 4067 (2022).
[9] Richards, D.M., Tweardy, M.J., Steinhubl, S.R. et al. Wearable sensor derived decompensation index for continuous remote monitoring of COVID-19 diagnosed patients. npj Digit. Med. 4, 155 (2021).
[10] Ketcheson, S.J., Golubev, V., Illing, D. et al. Application and performance of a Low Power Wide Area Sensor Network for distributed remote hydrological measurements. Sci Rep 13, 18050 (2023).”

(2) Table II should include a comparison of Bit Error Rate (BER) for each design, and the design itself should be benchmarked against others fabricated using die package methods.

[Reply]: Thank you for the constructive comment. In general, for the state-of-the-art OOK receivers, BER is not directly provided for comparison. It is provided along with the sensitivity performance at a particular standard BER of 0.001. All the works incorporated in the comparison table are based on the measurement results of the fabricated die.

[Modifications in the revised manuscript]:

1. (In pg. 6, Table II): Added BER in table II for the mentioned receiver sensitivity.
2. (In pg. 5, line. 166, text included and modified) “~~The bit-error rate (BER) performance of the proposed receiver is illustrated in Fig. 1(d). The sensitivity performance of the proposed receiver is determined as the input signal power level for which the bit-error rate (BER) is below 1e-3 as illustrated in Fig. 2(b).~~”

TABLE II
State-of-the-art OOK receiver performance comparison with the proposed receiver

	[7] JSSC'09	[8] ISSCC'10	[16] JSSC'18	[17] JSSC'19	[20] JSSC'21	[21] JSSC'21	This work	
Technology [nm]	90	90	14	45	65	65	40	
Architecture	Mixer-first Uncertain-IF	Detector-first	Mixer- first	Low-IF	Mixer-first, N-path filtering	Mixer-based Zero second IF	Mixer first Low-IF	
External components	BAW resonator	Inductors, Clocks	32 kHz RTC	2 MHz reference	FBAR, XO, Input matching	Input matching	None	
LO generation	Unlocked RO	--	Ring with FLL	LC-VCO	RO + PLL	RO + PLL + Frequency Tripler	Unlocked RO	
Modulation	OOK	OOK	OOK	FSK	OOK	OOK	OOK	
Power supply [V]	0.5	0.5	0.95	1/0.9	0.5	0.5	1/0.9	1/0.9/0.7
Frequency [GHz]	2	2.4	2.4	2.4	2.4	2.4	2.4	
Power [μ W]	52	51	95	1200	220	352	178	119
Latency [μ s]	N.R.	N.R.	6 ^a	N.R.	200 ^a	1470 ^a	0.295^b	
Data rate [kbps]	100	100	62.5	250	4	5	5000	1000
Sensitivity @ 1e-3 BER [dBm]	-72	-64	-72	-82.2	-85	-92	-62	-67
Area [mm ²]	0.1 [*]	0.36 ^{**}	0.19 ^{**}	1 ^{**}	1.3 ^{**}	0.6 ^{**}	0.4^{**}	
Energy efficiency [nJ/bit]	0.52	0.51	1.52	4.8	55	70.4	0.0356	0.119

(3) For instance, in VCRO section, crucial information such as phase noise value, mixer isolation, and overall design power efficiency are prominently absent.

VCRO section:

[Reply]: Thank you for the valuable comment. Since for the OOK receivers, the phase noise is not a critical parameter and to achieve an ultra-low power consumption, we have chosen the unique VCRO topology which is suitable for providing a considerably stable rail-to-rail LO signal against PVT. This information is updated as follows.

[Modifications in the revised manuscript]:

(In pg. 10, line. 329, text included) “In the proposed solution as a heterodyne OOK receiver, the phase noise of the VCRO is not critical [13] since the IF frequency can be uncertain and lie within the supported IF bandwidth for decoding amplitude modulated signals. Thus, the proposed VCRO topology is chosen to provide a stable LO signal with ultra low power consumption.”

Mixer isolation:

[Reply]: The VCRO output is AC-coupled to the mixer since the oscillator signal is rail-to-rail and there is no 50 Ω matching. Since our primary goal is to achieve good performance at very low power consumption, the LO is chosen to be generated as a rail-to-rail signal and drive the mixer through AC-coupling. RF to IF leakage is very

small due to 40dB/decade roll-off from the mixer conversion gain plot (Fig. 3(c)). LO to IF leakage is also small due to cascaded structure of the mixer topology. The high frequency components at the mixer output are eventually attenuated by the additional 60dB/decade roll-off from the 3 stage IF amplifier (Fig. 5(d)).

(c)

Fig. 3. Proposed pseudo-differential active mixer (c) Simulated voltage conversion gain.

(d)

Fig. 5(d) Simulated bandpass voltage gain characteristics of IF amplifier.

[Modifications in the revised manuscript]:

(In pg. 9, line. 295, text included) “The RF-IF and LO-IF isolation is very large as the RF input alone is matched to 50Ω. The LO port and IF port are AC-coupled and low-pass filtered respectively thus attenuating the RF and LO signals at the mixer output and further attenuated by the IF amplifier.”

Overall design power efficiency:

[Reply]: An equivalent information regarding the overall design power efficiency is provided in most of the state-of-the-art receivers [13,14,22,23,26,27] as a function of energy efficiency. As stated in the references [13,14,22,23,26,27], the energy efficiency is defined as the ratio of the overall receiver DC power consumption and the supported maximum data rate [18]. This information is provided in the original manuscript as a performance parameter “Energy efficiency” in the Table II.

(4) *The fabrication process details, particularly those related to transistor-based circuits, necessitate a more comprehensive explanation.*

[Reply]: Thank you for the valuable comment. The integrated circuit is designed by the authors, and the fabrication is performed in TSMC 40nm MS/RF CMOS process. A more detailed information on the fabrication process is provided and updated in the manuscript as follows.

[Modifications in the revised manuscript]:

(In pg. 3, line. 109, text included) “The proposed receiver design is fabricated in TSMC 40nm low power (LP) mixed signal/radiofrequency (MS/RF) CMOS process. The semiconductor fabrication process comprises 10 metal layers for inter/intra-device connections. Different types of transistors such as regular- V_T , low- V_T , and high- V_T devices are used in the receiver design. The high frequency circuits are designed using high-frequency RF transistors from the process design kit (PDK). These transistors possess foundry-provided optimized layout. The analog baseband circuits such as IF amplifier, envelope detector, comparator, and the baseband amplifier are designed using low-frequency transistors. The deep-nwell transistors are also available in the fabrication process which are partly utilized in the foundry-provided IO pad design. At each stage of the receiver design, appropriate transistors and other devices are chosen for optimum performance in the best knowledge of the authors.”

(5) Furthermore, the paper could benefit significantly from the inclusion of a software-based layout graphic to enhance the clarity of the presentation.

[Reply]: Thank you for the constructive comment. The manuscript is written using Microsoft Word format based journal’s template. We have improved the manuscript layout manually to enhance the clarity of the presentation. Figures are split and the paper layout is modified such that the clarity and connection of the text to the presented figures is improved. Display items (tables, figures) are appropriately positioned w.r.t the corresponding text. The number of allowed display items is 10 provided in the journal guidelines here.

[https://www.nature.com/commseng/submit/submission-guidelines#:~:text=Article%20formatting%20guidelines%3A%20an%20abstract,items%20\(figures%2C%20tables\).](https://www.nature.com/commseng/submit/submission-guidelines#:~:text=Article%20formatting%20guidelines%3A%20an%20abstract,items%20(figures%2C%20tables).)

[Modifications in the revised manuscript]:

1. Updated number of figures to 8 with constructive mapping of the text.
2. Appropriately positioned the display items in the paper layout.
3. Text accordingly modified in numerous places for changes in figure numbers.

(a)

(b)

Fig. 1. Proposed receiver design and experimental setup. (a) Receiver architecture block diagram. (b) Fabricated die microphotograph. (c) Setup schematic for the receiver measurements. (d) Test equipment – lab testbench.

Fig. 2. Proposed receiver design performance. (a) Simulated and measured data of input matching $|S_{11}|$. (b) Measured BER vs input signal power level. (c) Measured BER vs input signal frequency. (d) 5Mbps OOK

decoding in normal mode. (e) 1Mbps OOK decoding in low-power mode. (f) Latency measurement of the proposed receiver.

(c)

Fig. 3. Proposed pseudo-differential active mixer (a) Simplified schematic. (b) Full schematic in detail. (c) Simulated voltage conversion gain.

Fig. 4. Proposed voltage-controlled ring oscillator. (a) Schematic. (b) Simulated variation of output resistance of NMOS and PMOS transistors in saturation region w.r.t temperature. (c) Simulated frequency of VCRO output signal w.r.t temperature. (d) Simulated VCRO frequency tuning w.r.t V_{DD} .

Fig. 5. IF and baseband design blocks. (a) Schematic of the proposed IF amplifier with bandpass filter characteristics. (b) Variable resistor symbol. (c) Equivalent circuit schematic of the variable resistor. (d) Simulated bandpass voltage gain characteristics of IF amplifier.

(c)

(d)

Fig. 6. IF and baseband design blocks. (a) Effect of supply voltage on gate-bias voltage (V_g) and combined effective resistance of M_{N0} and M_{P0} . (b) Effect of temperature on gate-bias voltage (V_g) and combined effective resistance of M_{N0} and M_{P0} . (c) Voltage gain robustness with temperature variation. (d) Voltage gain robustness with supply voltage variation.

Fig. 7. IF and baseband design blocks. (a) Schematic of the implemented source-follower envelope detector. (b) Schematic of the implemented comparator block.

Fig. 8. Schematic of the proposed baseband amplifier.

REVIEWERS' COMMENTS:

Reviewer #3 (Remarks to the Author):

Authors replied to all the raised comments. I have no more comments